# Using Smartphones to Locate Trapped Victims in Disasters

**DOI:** 10.3390/s22197502

**Published:** 2022-10-03

**Authors:** Yenpo Tai, Teng-To Yu

**Affiliations:** Department of Resources Engineering, National Cheng Kung University, Tainan 701, Taiwan

**Keywords:** smartphone, Bluetooth, locating, hazard, victims

## Abstract

Large and unaccounted numbers of victims in disasters, events, or fires are often trapped in buildings or debris, and must be located and rescued as soon as possible. This study transforms smartphones into indoor locating tools without extra modification or complicated program installation, considering smartphones are likely to be carried when disasters strike. The study creates a system that converts smartphones into a lifesaving tool for trapped victims and rescuers. This study employs the Bluetooth beacon in smartphones to send signals using its low power consumption feature. The signal could continue transmitting for rescuers to locate trapped victims for longer. Rescuers could use the Bluetooth function on a regular notebook computer to search such signals without any hardware implementation or modification, allowing them to locate and determine the position of many trapped victims simultaneously. Implementing this system will decrease the search and rescue team’s need to enter unsafe areas and increase their rescue speed, a critical factor for the survival of trapped victims. Furthermore, when disasters strike, the smartphone calling function might not work, and the trapped victim might be too weak to call for help. Thus, autoreply messages from victims’ smartphones could help them be located within a 2-m error, even if covered by fallen debris such as wood piles or tiles. This effort will increase the chance of finding trapped victims within the golden rescue hours and reduce the exposure time of search and rescue teams in unsafe environments.

## 1. Introduction

Earthquakes often strike in major cities around the seismic belt, with casualties increasing as population density increases over time [1]. Disasters such as earthquakes and volcanic ruptures cause damage and endanger victims worldwide; thus, search and rescue are an international concern. Collapsed buildings after bomb blasting, either from war or terrorist attacks, can also trap victims. These circumstances, whether natural or human-induced disasters, require indoor positioning and search and rescue efforts. Indoor positioning systems consist of Bluetooth, Wi-Fi, WLAN, or radar signal transmitters and receivers [2,3], often requiring multiple machines or sophisticated system integration. Though the available indoor positioning systems can locate trapped victims within tens of centimeters error, they often require vast electrical power to operate, thus reducing the standby time of smartphones. 

Since disaster can strike without notification, there is no telling where and when a future hazardous event may occur. For such reasons, any indoor positioning system that requires pre-installation is not suitable for the task of quickly recovering trapped victims. Another issue in locating trapped victims is the persistence and availability of the necessary system; the chances of finding trapped victims are proportional to these two features. Furthermore, system availability is decided by the price, special hardware, working environment, and training personnel. This study converted smartphones with Bluetooth and regular NB into an indoor trapped victim locating system to fulfill such requirements. This is beneficial because there is no requirement for on-site electricity, internet access, or auxiliary devices. The operating system is simplified so anyone with basic computer skills can handle it. The most notable contribution of this work is to remove the technical and financial barriers to trapped victim locating systems and simplify the steps to reduce computing time. With adequate NB and rescue personnel, the likelihood of locating trapped victims suffering from major catastrophic hazards as soon as possible increases. Therefore, the chance of saving lives could be vastly improved.

To locate trapped victims, traditional search and rescue teams are equipped with rescue dogs, life detectors, sonar life detectors, and listening devices [4]. In addition to these measures, other devices search for life signs by detecting the trapped victims’ movement sounds (including breathing, beating, and calling). Since this type of equipment is stored at fire stations, it may not be quickly accessed; even in a major earthquake disaster, such as the Chi-Chi earthquake in Taiwan, only one-quarter of the personnel in search and rescue teams had the necessary equipment to detect life signs [5].

Shortage of firefighting personnel due to limited funds and old equipment is a common obstacle that downgrades the efficiency of search and rescue worldwide. As a result of inadequate equipment, firefighters are exposed to danger for a longer time during disaster relief duty. Although many international search and rescue teams have advanced rescue equipment, the quantity is often limited. When disaster strikes, international teams need time to reach the scene, which can take days to arrive. According to statistical data from Japan, earthquake victims’ survival rate is 80% on the first day. It drops rapidly to 30% on the second day, and only 10% to 15% of victims would survive if found on the third day [6]. Therefore, the time spent on the rescue effort is a crucial factor in determining the survival rate of victims. 

Deploying more personnel and facilities is a common way to reduce rescue time. Fire brigades from various regions could assist in disaster relief if their equipment is adequate and suitable. This study proposes an affordable and effective rescue system using smartphones to replace expensive life-detecting equipment, thus, equipping sufficient personnel with a limited budget. People are so highly attached to their smartphones that they even put them by their beds at night [7]. With this study design, smartphones can automatically activate the Bluetooth beacon when a disaster strikes and then transform into a locating tool to aid rescue missions. The search and rescue personnel only need a regular notebook computer with Bluetooth to locate nearby victims by taking signal strength measurements at three locations. This system’s tens of centimeters accuracy could significantly reduce the needed sweep time of traditional victim search and rescue tactics. This study proposes an affordable, effective, and universal rescue system that could be in situ to deploy tens or hundreds more personnel than expensive life-detecting machines. The low power consumption of Bluetooth and the 10-m signal transmitting distance make it a perfect tool for the low-cost lifesaving tool of locating trapped victims.

## 2. Related Works

Search and rescue efforts often employ various helpful strategies. The joint effort of search dogs with wearable equipment and trained personnel can locate trapped victims via the position returned from the dog’s equipment [8]. However, the system stability is affected by the training of dogs or complex environmental factors. Therefore, it is not a consistent system under various external conditions, which does not offer qualified reliability. Improvements can be made to existing search and rescue efforts using wireless signals, such as smartphone networks; for instance, carrying small smartphone base stations on drones and using mini base stations to locate the smartphones of trapped victims [9]. However, this design not only increases the search and rescue cost but also depends on drones to find the smartphone wireless signal of a trapped person that requires the GPS or Wi-Fi signals to pinpoint victims [10]. Some employ drones to find the Wi-Fi signal associated with GPS for locating [11] and use the Wi-Fi signal of UAVs for transmitting information [12]. 

However, these methods are not designed for indoor search and rescue, considering that smartphone base stations might not work properly at the disaster site. Further, GPS signal is usually poor indoors. Another limitation of the current technologies employed is when rescue systems use drones with Bluetooth Low Energy (BLE): operators must wear a BLE device to operate the system [13]. Since disaster areas often have affected environments, the network connections may be unstable in many cases. Therefore, specific client-server architecture methods are not always reliable in disaster areas. Although some solutions use mobile phones and Bluetooth [14] to design the rescue system, the network system in the disaster area is often unstable. Some research will first deploy the Bluetooth beacon, Wi-Fi, or other sensors in the building [15,16,17,18,19,20] and use the signal strength of the sensor between the cell phone/receiver to estimate the distance. Then, the cell phone or receiver transmits signal data to the server for calculation. However, this method is challenging to employ in a disaster area environment. Strong earthquakes can cause buildings to collapse, and pre-deployed sensors will deviate from their original position; relative positions obtained could be problematic in transmitting signals. Another drawback is that networks might not work after an earthquake for the mobile phone or receivers to send the signal to the server. Therefore, indoor pre-deployed sensors are not suitable to rely on during a post-earthquake environment. There are also ways to use non-wireless signals, such as using lasers and robots to construct positioning maps [21] or using sound to locate trapped people [22]. All the above systems are expansive and require special training, which is unlikely to be deployed to all rescue units. An occupancy detection method that uses BLE smartphone devices to perform zone-level occupant localization. The proposed method uses a network of BLE beacons to record the received signal strength indicator (RSSI) values of neighboring devices, which were consolidated and pre-processed to obtain a set of RSSI tuples [23]. Occupation profile and density could be gathered if the building is equipped with sensors and routers. This could provide the pre-event information for the quantities of occupants that have been trapped inside the building.

Many life detectors are commercially available products and have different principles, such as imaging equipment with optical fibers to connect the camera and display screen [24]. With the probe pole, the image can be used to find the trapped person, but the disadvantage of detection distance is limited by the length of the probe pole, about 2.4 to 3.7 m. The vibration sensor that detects breathing, tapping, and calling signals to identify the victim’s location is also available [25]. This sensor is categorized into two types: one is an eight-meter short-distance wired connection, and the other is a 100-m wireless connection.

A disadvantage of the vibration-type life detector is that disaster sites have many vibrations, interfering with the signals and affecting detection accuracy. Another issue is that the batteries of life detectors typically last for only 2.5 h, and it takes 3.5 h to charge them. Carrying more battery sets around disaster sites is not a practical option because of crowded spaces and the carrying weight limit of rescue personnel.

In this study’s proposed system, the BLE beacon can be passively activated if the victim has pre-installed the required app. The most significant advantage of the BLE beacon is its low power consumption, which can prolong the system’s operation and increase the probability of finding trapped victims. In addition, the rescue personnel only need a regular notebook computer (NB) instead of special gear. Ultimately, it is an affordable and effective system that could significantly deploy many rescue personnel to improve their search capacities.

## 3. Localization Method

This study’s designed method allows an individual to estimate the location of a trapped person via triangulation by locating the BLE beacon signal source. The system architecture turns the smartphone of the trapped person into a beacon that can send the signal. Then, using a notebook computer to measure the signal strength from the beacon at three locations, the trapped person’s position can be found according to the signal attenuation with increased distance. The detailed function of an individual unit is described as follows.

### 3.1. Signal Source of Cell Phone

Different signals on smartphones can be used to attract the attention of search and rescue personnel, such as the smartphone signal, voice, and GPS. During indoor search missions, it is necessary to have a search and rescue system that relies on the existing signals of smartphones. The signals suitable for search and rescue targets are Wi-Fi, NFC, and BLE [26]. The transmission distance of the NFC signal is too short, only around 10 cm. On the other hand, while the Wi-Fi signal coverage is 100 m, its power consumption is too high, shortening the functional time of a victim’s smartphone and reducing the available rescue time. The BLE signal covers a range of 10–50 m, and it has a low power consumption that makes it suitable for search and rescue missions, considering the searching distance and system endurance.

The Beacon simulator application transformed the iPhone 11 into a BLE beacon in our experiment. A power consumption measurement for the smartphone in the disaster environment was performed to examine the BLE power consumption on the smartphone. When the Beacon simulator application is turned on, the phone switches to flight mode and turns off 4G and Wi-Fi. The smartphone screen must remain on and cannot enter the auto-off mode while the Beacon simulator application is in use. We found that the iPhone 11 consumes about 2% power per hour. After four hours of testing, the total power consumption was around 9%. If a smartphone’s battery is fully charged, it can support more than 40 h of such operation, greatly extending the likelihood of finding a trapped person.

### 3.2. Detect System on Notebook Computers

Once the BLE beacon is selected as the signal to detect a target, a compatible detection and computing platform must be established. Since the environment of the search and rescue scene changes over time and place, the size of the search and rescue system should preferably be light, thin, and short in size. The system includes BLE signal measurement, signal classification, and locating function components, considering that the BLE signal measurement function needs to obtain signals at three locations in different disaster environments.

The existing locating processing parameters must be adjusted according to the situation. It requires working with the original Received Signal Strength Indicator (RSSI) signal [27]. Through offline training and online filtering, the locating errors were about 1.5 m [28]. While this application installed on the smartphone can obtain the estimated distances from beacon signals, further data processing is beyond the capability of current smartphones. In addition, the position estimation function also requires advanced computing routines, such as MATLAB or Python. It requires advance computing skills with a battery that provides adequate endurance and pre-installed special software, making it impossible to requisition general NB for signal detecting at disaster sites. The proposed search and rescue operations can be accomplished on a regular NB without needing to access the server system via the internet since it cannot guarantee internet access to upload data for cloud computing during a search and rescue scenario.

The Bluetooth signal detection module was rewritten from the Universal Beacon Library [29] to record the BLE signal strength since the library does not provide a long-term logging function.

The Bluetooth module analyzes the collected signal and handles the individual environmental influence on the Bluetooth signal. Figure 1 shows four signals at different distances that reveal various signal strengths. The distance estimated formula based on the Bluetooth signal is shown below.
(1)$$d=10^{\frac{\mathrm{RSSI}\left(d0\right)-\mathrm{RSSI}\left(d\right)}{10n}}$$

Measure distance is denoted as d, RSSI(d) is the referencing signal strength in an environment, and the RSSI(d0) is the Bluetooth signal strength measured at nearby locations. The value of n has different effects due to environmental factors [30]. Therefore, when establishing a system, it is necessary to confirm whether the measurements are located within the same level and verify whether the measurements perform better at the vertical or horizontal plane to reduce possible locating errors.

A drawing of the measured signal strength at different points with vertical separating intervals is presented in Figure 2. The other datasets measured at the horizontal intervals are shown in Figure 3. We found that the horizontal measurement signal was relatively stable compared to the vertical counterpart. The numerical analysis found a possible significant locating error in the vertical than the horizontal plane, as shown in Table 1. For the same amount of separation of 200 cm, the dBm changing range is 18~22 dBm at the vertical plane and 10~18 dBm at the horizontal plane.

The Bluetooth signal was carried out through sweep frequency; the recorded signal fluctuated randomly. There were two groups of signal strength within the signal measurements—one group had stronger signal strength, and the other was weaker. The strength of recorded signals is illustrated in Figure 4, showing the two randomly caused distinct signal strengths.

The Bluetooth ranging theory dedicates the possible distance from the radio wave attenuation through formula conversion. However, Bluetooth is carried out by sweeping the frequency, so the tested object and the observation point might have diverse radio waves and travel paths within a disaster scenario. If one uses all the collected points to calculate the distances of the test object to the observation point, there will be too many combinations. Therefore, the obtained distance was measured from the centroid of the strong or weak signal group. As shown in Figure 5, the signal strength was separated into two categories after the k mean grouping method. The two centroids of these two categories represent the strongest and weakest signals from the observation point.

The complicated situation of disaster sites leaves room for measured distance biases via the Bluetooth signal decay method. For one, the N value of the Bluetooth estimation formula of Equation (1) will change due to environmental alterations. The measured distance biases at the same observation point are caused by the disturbed signal and have about a meter of inaccuracy. As shown in Table 2, the N value was adjusted to improve the accuracy of the estimated distance. Then, the error was reduced to tens of centimeters via this adjustment.

With one distance measurement from the target located within a circle, the trilateral measurement method at three not collinear points can offer the target point solution [31]. The uncertainty of the disaster environment and the sensitivity of the Bluetooth signal makes the absolute position of objects impossible to obtain. Therefore, the wave signals must be observed at three randomly selected points in the disaster site to overcome such a deficit. Then, the ‘findminsearch’ function of MATLAB is applied to estimate the position with the minimum solution matches.

## 4. Results

The system was validated by simulating different disaster environments by measuring signal decay and fluctuation. The experimental condition was to place a mobile phone in the coordinate axis system unit in meters. The phone was placed at the position of (x, y, z) = (0, 6.4, 0). The simulated observation positions of the search and rescue personnel were at (2.4, 2.4, 1) (0, 0, 1), and (2.4, 0, 1), respectively. There are no obstructions around the smartphone signal transmitter in the first situation. In the second test, the source of the smartphone signal is shielded by the wooden board, as shown in Figure 6. The third situation was done by shielding the source of the smartphone signal with tiles, as shown in Figure 7.

After the calculation, the possible positions of the trapped persons under different shielding conditions were obtained. The following figures denote various combinations; the red, blue, and green circles are the three observation points (0, 0, 1), (2.4, 0, 1), and (2.4, 2.4, 1). Figure 8 shows the testing result without any source signal breakage. The estimated error is around 4.5 m to 2.4 m. The results of the source signals blocked by wood are illustrated in Figure 9; the estimated locating error is about 5.1 to 2 m in this case. Figure 10 denotes that tiles have blocked the source signals, and the estimated locating error is also around 2.9 to 2.4 m. Therefore, it is evident that the system can indeed exert the ability of rapid search and rescue within a feasible range—an alternative to conducting a ground-based search at a disaster site. The results were acquired while the observation points were not evenly spread around the target. If improved observation geometry could be evenly distributed, then the accuracy of locating is around 1 m.

## 5. Conclusions

Existing passive signal life-detecting systems are often too expansive or require special training, making their universal deployment impossible, especially within non-wealthy counties. Smartphones have become an essential part of modern life. When disasters strike, there is a high probability that trapped victims will be carrying their smartphones compared to other technology. In this study, the established search and rescue system could quickly alter smartphones into beacon emitters. By converging the Bluetooth signal beacon into the needed ranges for rescuers, they can reduce the search time and improve the chance of finding victims in time. Based on the simulation test results, this system could locate victims within a two meter-error even in the worst condition, such as under wood piles or tile blockages, and not evenly distributed observation points. The simplified algorithms need three measurements at various locations and take a few seconds to locate the trapped victim via entry-level NB without any extra hardware.

The design and operation of our system are straightforward; any person with basic computer skills could handle them by following the manual instruction. Any entry-level computer with Bluetooth could run this inexpensive solution. Thus, this system is a feasible solution requiring very little budget; most rescuers could be equipped with regular NB computers to expand their rescue capacities. In any historical major catastrophic event, not all rescuers are adequately equipped; the rescue’s golden time is spent on turning system deployment or waiting for shipping from overseas. The proposed system’s low cost and easy operating features could overcome previous drawbacks and find more trapped victims within the first 72 h of golden rescue time. Any volunteer civilian could use their own NB to install the proposed APP via USB stick, and then an extra detecting unit is established.

Furthermore, a Public Warning System (PWS) [32] in Taiwan sends early warning text messages indicating disaster. The received text message could be set to activate the smartphone Bluetooth function; it could immediately transform into a beacon emitter for the trapped victims. This will prevent people from forgetting to activate the Beacon function due to nervousness or panic and help trapped victims who cannot move or stay conscious. Since Bluetooth consumes little power from smartphones, the beacon could emit a signal for more than 48 h if fully charged, significantly improving the chance of finding trapped victims. In addition, trapped victims could also save physical strength without the need to shout and continuously call for help to attract the rescuers’ attention, thereby increasing the possibility of being rescued. Mobile phones can link to the health watch, and Bluetooth Beacon packets can send that specific information outwards. Since persons typically wear smartphones and watches, the trapped victim’s heartbeat and blood oxygen level could be remotely accessed. Rescuers could allocate priority to those people with low blood oxygen or shortness of breath to increase their survival chances.

## Figures and Tables

**Figure 1 sensors-22-07502-f001:**
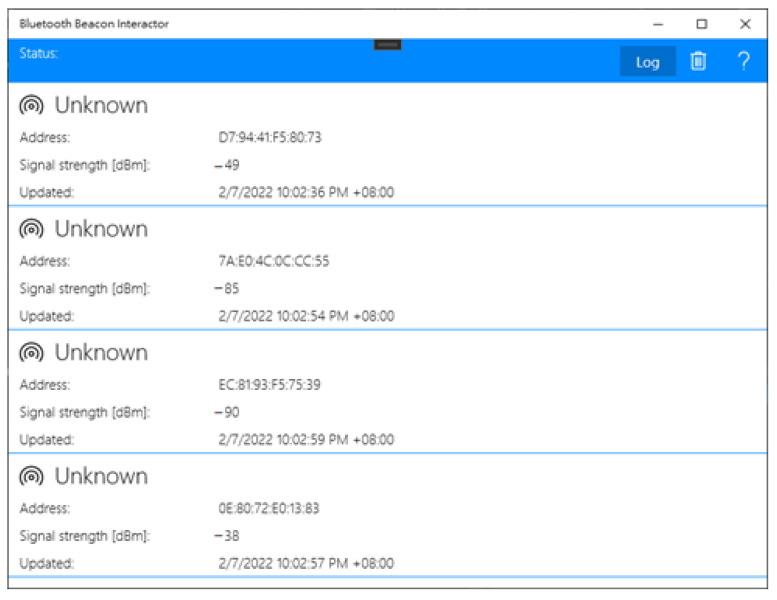
User Interface of Bluetooth signal captured App.

**Figure 2 sensors-22-07502-f002:**
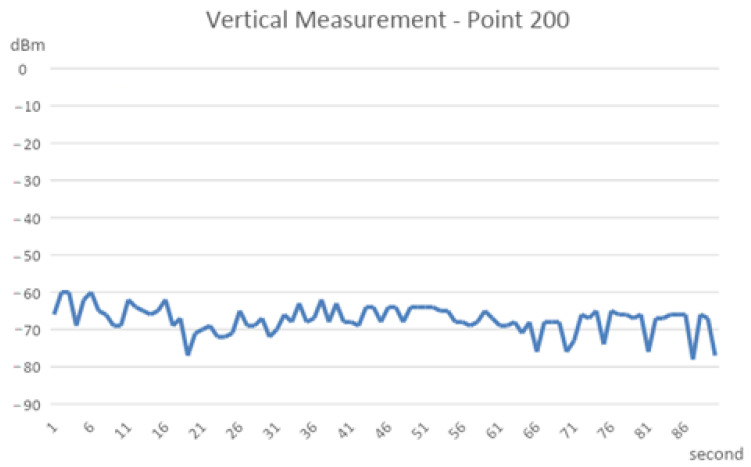
Line chart of the vertical measurement signal.

**Figure 3 sensors-22-07502-f003:**
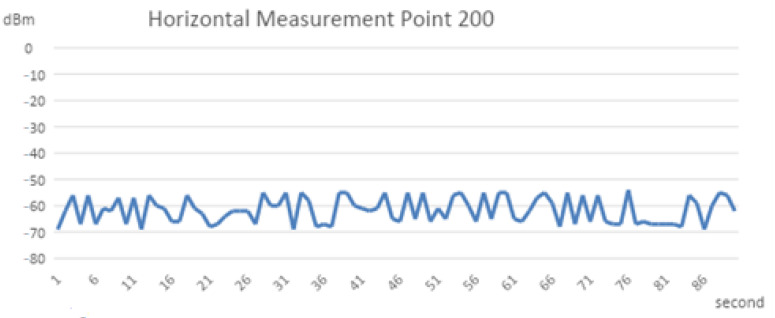
Line chart of the horizontal measurement signal.

**Figure 4 sensors-22-07502-f004:**
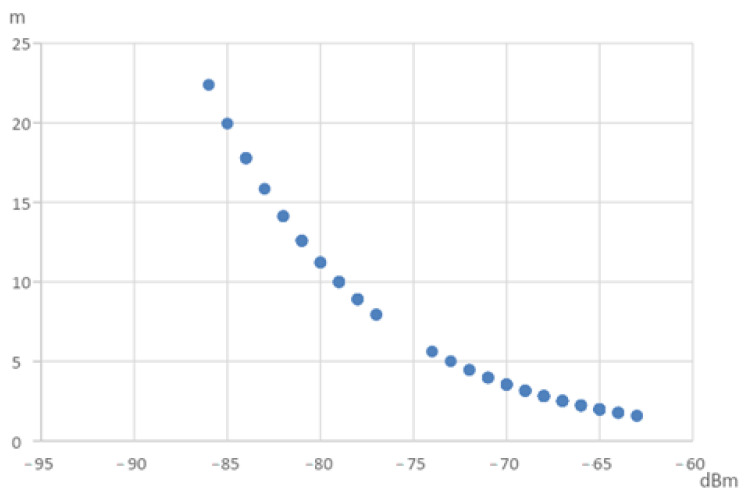
Bluetooth signal distribution diagram.

**Figure 5 sensors-22-07502-f005:**
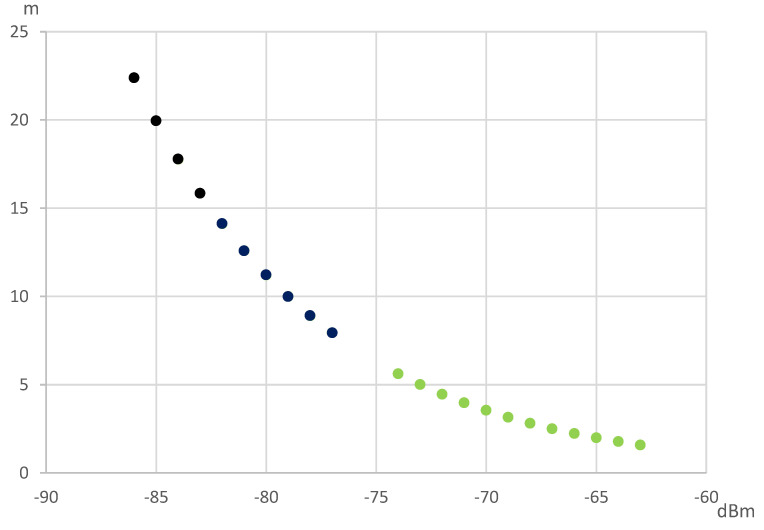
Two Bluetooth signal distribution groups after k mean clustering.

**Figure 6 sensors-22-07502-f006:**
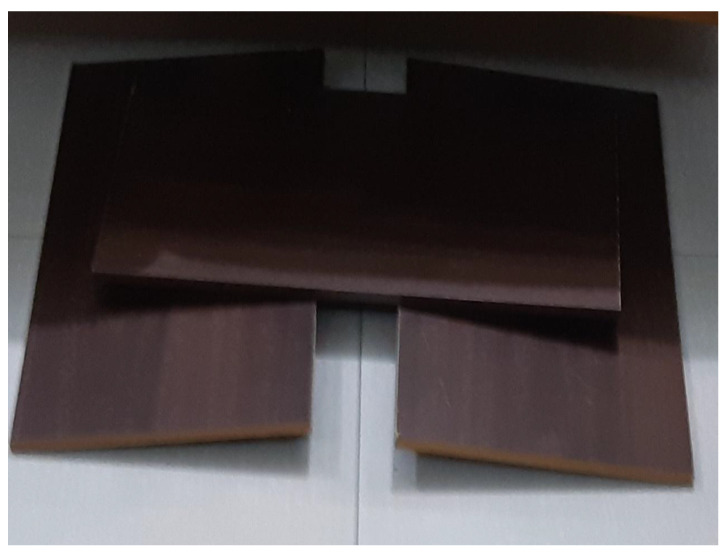
The experimental setup of shielded signals by wood.

**Figure 7 sensors-22-07502-f007:**
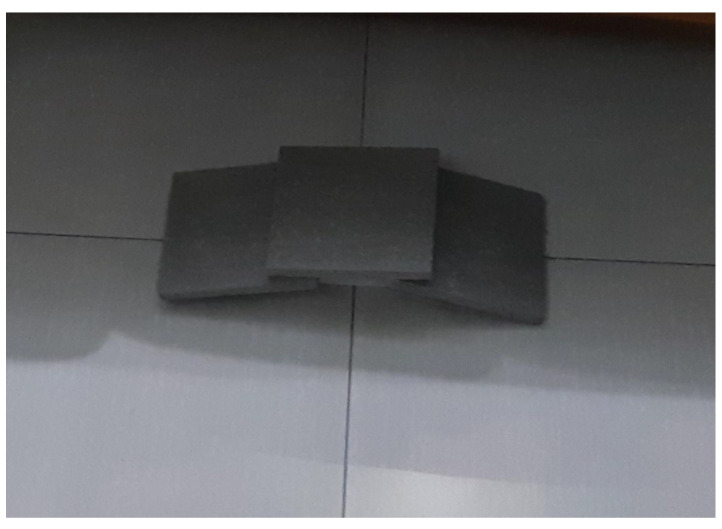
The experimental set of shielded signals by tiles.

**Figure 8 sensors-22-07502-f008:**
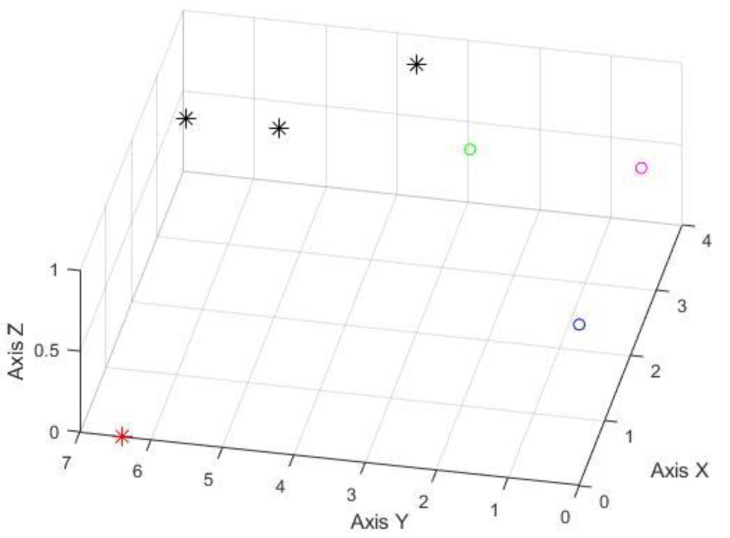
The experimental results of no shielded signal (unit in meter); the red asterisk represents the signal origin; the black asterisks are the estimated positions from three on-site testing. The red, blue, and green circles are the locations of three BLE signal observation points.

**Figure 9 sensors-22-07502-f009:**
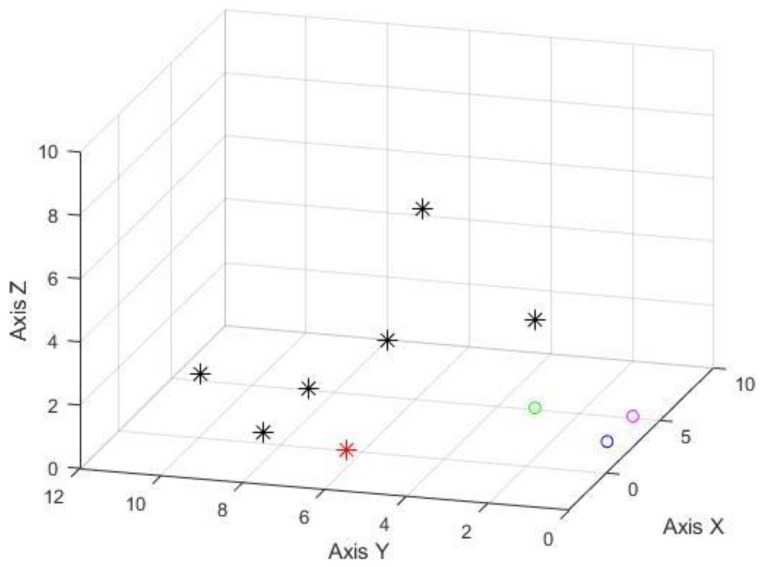
The experimental results of the shielded signal by wood (unit in meter), the presentation of symbols is the same as Figure 8.

**Figure 10 sensors-22-07502-f010:**
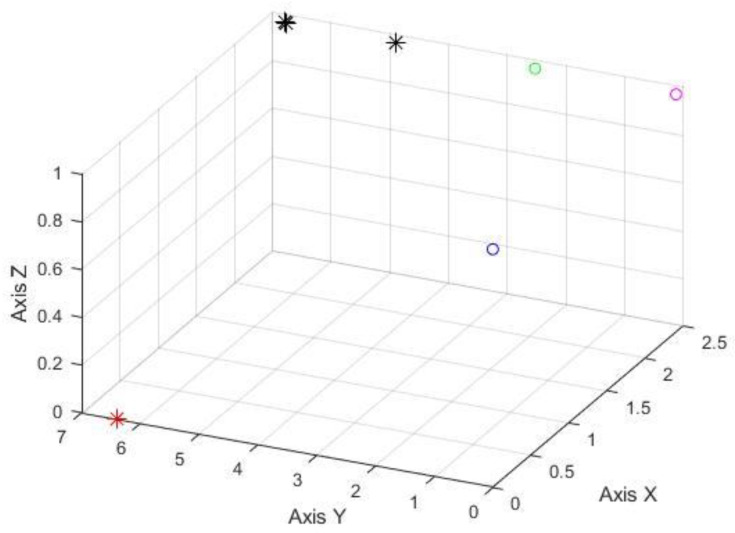
The experimental results of the shielded signal by tiles (unit in meter), the presentation of symbols is the same as Figure 8.

**Table 1 sensors-22-07502-t001:** Numerical analysis of various settings of Bluetooth signal.

	STD Value	Max Value	Min Value	Data Range
Vertical Interval of 200 cm	3.7377	−60	−78	18
Vertical Interval of 0 cm	3.4842	−61	−83	22
Horizontal Interval of 0 cm	2.4491	−55	−65	10
Horizontal Interval of 100 cm	3.7732	−51	−69	18
Horizontal Interval of 200 cm	4.9416	−54	−69	15

**Table 2 sensors-22-07502-t002:** The setting value of N for various wave value.

N	Wave Value
1	>−69
2	<−70 and >−79
3	<−80

## Data Availability

Not applicable.

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
