# Peer review of "Using Smartphones to Locate Trapped Victims in Disasters"

_sensors, 2022, doi:10.3390/s22197502_

Round 1

Reviewer 1 Report

The manuscript presents a novel approach to locate trapped victims in disasters using a mobile application that acts like a BLE Beacon when activated. Even though the proposed idea is interesting and worth developing, I believe that the manuscript and the performed research have a lot to improve. 

First, the manuscript has not been proof-read and divided into sections. In the introduction there is text from the journal template, there is no section 2 or related work section, it goes directly from Introduction section to results section, in line 86 there is an incomplete sentence, etc. Relevant related works are missing. For instance:

Fasano, L., Sergi, I., Almeida, A., Jayo, A. B., Rametta, P., & Patrono, L. (2020, September). Performance Evaluation of Indoor Positioning Systems based on Smartphone and Wearable Device. In 2020 5th International Conference on Smart and Sustainable Technologies (SpliTech) (pp. 1-5). IEEE.

In figures 8,9 and 9 it is not specified the meaning of the numbers, in the evaluation is not specified the specific conditions on which the metrics has been taken (distance to the phone for instance). A more in depth evaluation and discussion should be done.

Author Response

  1. we thank the reviewer for carefully poring over our manuscript and offers helpful comments which substantially improve our manuscript. We have reorganized some textual errors and omissions and related literature. The related work section has been added and in-depth analysis about this system construction and performance is also provided. Major contribution of this work is to cut down the system hardware requirement and simplified algorithms computation to save the handling procedures and time. It is proven that with any Bluetooth smart-phone and regular NB, this system could detect buried trapped victims within 10 meters radius in meter level accuracy. All the requiring steps are taking three measurements at various locations then using a simplified RSSI calculating method to locate the target. All the procedures could be performed within 5 minutes, if there is no obstacle in the working environment then all the tasks could be done in 3 minutes. The long-lasting feature of smart phone Bluetooth allow this system to run the carpet search for any possible trapped victims at early stage of the rescue effort.
  2. We very thank the reviewer for the literature provided, and we also included this article into the related work. The revised part is shown in italic font as follows:

    Indeed, some solution using mobile phones and Bluetooth for rescue system, this networked system requires individual RSSI calibration which is kind of time-consuming work and it take training personnel to handle it. With this obstacle, it is difficult to vastly deploy for rescue mission of major scale catastrophic event.

  3. Figures 8, 9, 10 show the x, y, z coordinates in the indoor space unit in meter, and we added x, y, z axes in figure 8, 9, 10. The asterisk indicates the location of mobile phone. We also added the coordinate position of the mobile phone and the distance range between the estimated point and mobile phone in the text.

Reviewer 2 Report

MDPI Sensors Journal (Manuscript ID: sensors-1918208)

Comments to the Author

This paper investigates the idea of using smartphones as indoor locating tool using Bluetooth function to locate trapped victims. It is an interesting topic and the paper studies the concept clearly. However, there are several points need to be addressed to improve the quality of the manuscript.

Suggestions to improve the quality of the paper are provided below:

1)     In the author’s name list both affiliations are the same therefore there is no need to include (2). Also only corresponding author’s email address is sufficient. So please remove the (2).

2)     Abstract needs to be structed better. Current version contains a very long problem statement which left no space to include details about the methodology and results. Please restructure the abstract in a more concise and informative manner.

3)    The authors should clearly state the novelty of the paper, which is currently missing from the manuscript. Also, the main contributions stated in the Introduction should be rephrased to more clearly highlight its novelty over the existing literature and not just a description of what was done.

4)     Some sections missing section numbers. For example after “1. Introduction” another section follows as “Localization Method”. Please ensure each section and subsection have the section numbers properly.

5)     Please include a literature review section which is currently combined together with the Introduction section . It is very hard to follow in this form so please separate them.

6)    I strongly suggest that the authors include a brief description of the different applications of indoor localisation to help to attract other researchers who are working on similar applications to be interested in your paper. For instance, in the building domain, many indoor localisation technologies have been proposed to enable interesting applications such as building energy management, occupancy detection and smart building controls. Please include the following papers to get an idea of these applications:

Indoor localisation for building emergency management

Filippoupolitis, A., Oliff, W., & Loukas, G. (2016, December). Bluetooth low energy based occupancy detection for emergency management. In 2016 15th international conference on ubiquitous computing and communications and 2016 International Symposium on Cyberspace and Security (IUCC-CSS) (pp. 31-38). IEEE.

Indoor localisation for smart plug load control

Tekler, Z.D., Low, R., Yuen, C. and Blessing, L., 2022. Plug-Mate: An IoT-based occupancy-driven plug load management system in smart buildings. Building and Environment, p.109472.

Indoor localisation for smart HVAC controls

Balaji, B., Xu, J., Nwokafor, A., Gupta, R. and Agarwal, Y., 2013, November. Sentinel: occupancy based HVAC actuation using existing WiFi infrastructure within commercial buildings. In Proceedings of the 11th ACM Conference on Embedded Networked Sensor Systems (pp. 1-14).

7)     Please provide a more thorough comparison between different indoor positioning technologies to provide a stronger justification on the advantages of using the Wifi technology. For instance, the Bluetooth Low Energy (BLE) technology is viewed as a more energy efficient alternative to the Wifi technology as described in the following papers. Please review and include these below:

Tekler, Z.D., Low, R., Gunay, B., Andersen, R.K. and Blessing, L., 2020. A scalable Bluetooth Low Energy approach to identify occupancy patterns and profiles in office spaces. Building and Environment171, p.106681.

Filippoupolitis, A., Oliff, W. and Loukas, G., 2016, October. Occupancy detection for building emergency management using BLE beacons. In International Symposium on Computer and Information Sciences (pp. 233-240). Springer, Cham.

Other wireless technologies such as RFID has also been adopted by other studies due to its low cost and high localisation accuracy as described in the following papers:

Hahnel, Dirk, et al. "Mapping and localization with RFID technology." IEEE International Conference on Robotics and Automation, 2004. Proceedings. ICRA'04. 2004. Vol. 1. IEEE, 2004.

Li, N. and Becerik-Gerber, B., 2011. Performance-based evaluation of RFID-based indoor location sensing solutions for the built environment. Advanced Engineering Informatics25(3), pp.535-546.

8) Section 5. Patents seems empty. Please remove if you will not share any information about the patent.

9) Finally, in the conclusion section, there should be potential research directions regarding the proposed application. Please include a paragraph that could be useful for researchers who would like to benefit from this application.

Author Response

  1. we thank the reviewer for carefully poring over our manuscript. We follow the reviewer's comments and removed the affiliation number (2)
  2. According to the reviewer's comments, we shorted the problem description, and revise the abstract.

  3. Thanks to the reviewer for focusing on the novelty part. We has add a new paragraph in Introduction to state the novelty of the paper and the paragraph is shown as following.

    “Since disaster will stroke without any notification, there is no telling about where and when for the future hazardous event. For such reason, any indoor positioning system that requires pre-installation is not suitable for the task. Another issue regarding to the trapped victims locating is the persistence and availability of the system, chances of finding trapped victims are proportional to these two features. System availability is decided by the price, special hardware, working environment and training personnel, this study convert smartphone with Bluetooth and regular NB into an indoor trapped victim locating system to fulfill such requirement. There is no requirement for on-site electricity, internet access or any kind of auxiliary devices, and the system operating is simplified that any person with basic computer skill could handle. The most special contribution of this work is to remove the technical and finance barriers for trapped victim locating system, time consuming steps are simplified to reduce the computing time. If there is adequate NB and rescue personnel, then many trapped victims suffered from major catastrophic hazard could be located as soon as possible. Therefore, the chance of living could be vastly increased.”
  4. Thanks for the reviewer's correction, we has added the section numbers and make sure the numbers are in place in the manuscript.

  5. We has follow the review comments and separate the literature review section from the introduction section. In the revised version of the manuscript, Section 1 is the introduction and Section 2 is the literature review.

  6. We very thank the reviewer for the literature provided, and we has also included those articles into the related works. The revised part is shown in italic font as follows:

    “Some research will first deploy the Bluetooth beacon, Wi-Fi or other sensor in the building [15][16][17][18][19][20] and use the signal strength of sensor between the mobile phone or receiver to estimate the distance, then cell phone or receiver transmit signal data to the server for calculation. However, those method is very difficult to have such an environment in the disaster relief field. Strong earthquake will cause the building to collapse, so those pre-deploy sensors will deviate from the original position, so some relative positions obtained will be problematic. Another drawback is network might not work after earthquake for the mobile phone or receivers to send the signal to the server. The Therefore, those pre-deploy sensors in door positioning solution are not suitable for the post-earthquake environment.”

  7. We thank the reviewer for the literature provided. In Section 2, we elaborate different indoor positioning technologies and also included the articles into the related works. The revised part is shown in italic font as follows:

    “Some research will first deploy the Bluetooth beacon, Wi-Fi or other sensor in the building [15][16][17][18][19][20] and use the signal strength of sensor between the mobile phone or receiver to estimate the distance, then cell phone or receiver transmit signal data to the server for calculation. However, those method is very difficult to have such an environment in the disaster relief field. Strong earthquake will cause the building to collapse, so those pre-deploy sensors will deviate from the original position, so some relative positions obtained will be problematic. Another drawback is network might not work after earthquake for the mobile phone or receivers to send the signal to the server. The Therefore, those pre-deploy sensors in door positioning solution are not suitable for the post-earthquake environment.”

  8. Thanks for the reviewer's correction. The patent section is removed.
  9. we added the potential research directions in the conclusion section. The revised part is shown in italic font as follows.

     “At present, mobile phones could link to the health watch and Bluetooth Beacon packets can send those specific information outwards. While smartphone and watch are attached, the heartbeat and blood oxygen level of the trapped victim could be remote accessed. Rescuers could arrange the priority to those people with low blood oxygen or shortness of breath to increase their survival chance.

Round 2

Reviewer 1 Report

The quality of the paper has improved. However, please add to the camption of the figures what the asterisks of different color mean. 

Author Response

We have modified the figure caption as following:

Figure 8. The experimental results of no shielded signal (unit in meter); the red asterisk represents the signal origin; the black asterisks are the estimated positions from three on-site testing. The red, blue, and green circles are the locations of three BLE signal observation points.

Figure 9. The experimental results of the shielded signal by wood (unit in meter), the presentation of symbols is the same as figure 8.

Figure 10. The experimental results of the shielded signal by tiles (unit in meter) , the presentation of symbols is the same as figure 8.

Reviewer 2 Report

While the authors have addressed most of my comments and concerns, regarding the literature review, they have missed one of the references I have highlighted in the reference list.  Please include it in the manuscript to make the literature review complete.

For the application of BLE technology,

Tekler, Z.D., Low, R., Gunay, B., Andersen, R.K. and Blessing, L., 2020. A scalable Bluetooth Low Energy approach to identify occupancy patterns and profiles in office spaces. Building and Environment, 171, p.106681.

After addressing this, the manuscript is acceptable. Overall, great work!

Author Response

We have added this reference as [23] and introduce it as following: from line 121~127.

An occupancy detection method that using BLE smartphone devices to perform zone-level occupant localization. The proposed method uses a network of BLE beacons to record the received signal strength indicator (RSSI) values of neighboring devices which were consolidated and pre-processed to obtain a set of RSSI tuples [23]. Occupation profile and density could be gathered if the building equipped with sensor and routers. This could provide the pre-event information for the quantities of occupant been trapped inside the building.